# Transition to Linearity of Wide Neural Networks is an Emerging Property of Assembling Weak Models

**Chaoyue Liu**
Depart. of Computer Science
Ohio State University
liu.2656@osu.edu

**Libin Zhu**
Depart. of Computer Science
UC, San Diego
l5zhu@ucsd.edu

**Mikhail Belkin**
HDSI
UC, San Diego
mbelkin@ucsd.edu

## Abstract

Wide neural networks with linear output layer have been shown to be near-linear, and to have near-constant neural tangent kernel (NTK), in a region containing the optimization path of gradient descent. These findings seem counter-intuitive since in general neural networks are highly complex models. Why does a linear structure emerge when the networks become wide? In this work, we provide a new perspective on this "transition to linearity" by considering a neural network as an assembly model recursively built from a set of sub-models corresponding to individual neurons. In this view, we show that the linearity of wide neural networks is, in fact, an emerging property of assembling a large number of diverse "weak" sub-models, none of which dominate the assembly.

## 1 Introduction

Success of gradient descent methods for optimizing neural networks, which generally correspond to highly non-convex loss functions, has long been a challenge to theoretical analysis. A series of recent works including Du et al. (2018; 2019); Allen-Zhu et al. (2019); Zou et al. (2018); Oymak & Soltanolkotabi (2020) showed that convergence can indeed be shown for certain types of wide networks. Remarkably, Jacot et al. (2018) demonstrated that, when the network width goes to infinity, the Neural Tangent Kernel (NTK) of the network becomes constant during training with gradient flow, a continuous time limit of gradient descent. Based on that Lee et al. (2019) showed that the training dynamics of gradient flow in the space of parameters for the infinite width network is equivalent to that of a linear model.

As discussed in (Liu et al., 2020), the constancy of the Neural Tangent Kernel stems from the fact that wide neural networks with linear output layer "transition to linearity" as their width increases to infinity. The network becomes progressively more linear in $O(1)$-neighborhoods around the network initialization, as the network width grows. Specifically, consider the neural network as a function $f(\theta)$ of its parameters $\theta$ and write the Taylor expansion with the Lagrange remainder term:

$$f(\theta) = f(\theta_0) + \nabla f(\theta_0)^T (\theta - \theta_0) + \frac{1}{2}(\theta - \theta_0)^T H(\xi)(\theta - \theta_0), \tag{1}$$

where $H$ is the Hessian, the second derivative matrix of $f$, and $\xi$ is a point between $\theta$ and $\theta_0$. The "transition to linearity" is the phenomenon when the quadratic term $\frac{1}{2}(\theta - \theta_0)^T H(\xi)(\theta - \theta_0)$ tends to zero in an $O(1)$ ball around $\theta_0$ as the network width increases to infinity.

In particular, transition to linearity provides a method for showing that the square loss function of wide neural networks satisfy the Polyak-Lojasiewicz (PL) inequality which guarantees convergence of (Stochastic) Gradient Descent to a global minimum (Liu et al., 2022).

While the existing analyses demonstrate transition to linearity mathematically, the underlying structure of that phenomenon does not appear to be fully clear. What structural properties account for it? Are they specific to neural networks, or also apply to more general models?

In this paper, we aim to explain this phenomenon from a more structural point of view and to reveal some underlying mechanisms. We provide a new perspective in which a neural network, as well as

each of its hidden layer neurons before activation, can be viewed as an "assembly model", built up by linearly assembling a set of sub-models, corresponding to the neurons from the previous layer. Specifically, a $(l + 1)$-layer pre-activation neuron

$$\tilde{\alpha}^{(l+1)} = \frac{1}{\sqrt{m}} \sum_{i=1}^{m} w_i \alpha_i^{(l)},$$

as a linear combination of the $l$-th layer neurons $\alpha_i^{(l)}$, is considered as an assembly model, whereas the $l$-layer neurons $\alpha_i^{(l)}$ are considered as sub-models. Furthermore, each pre-activation $\tilde{\alpha}_i^{(l)}$ is also an assembly model constructed from $(l-1)$-th layer neurons $\alpha_i^{(l-1)}$. In this sense, the neural network is considered as a multi-level assembly model. To show $O(1)$-neighborhood linearity, we prove that the quadratic term in the Taylor expansion Eq.(1) vanishes in $O(1)$-neighborhoods of network initialization, as a consequence of assembling sufficiently many sub-models .

To illustrate the idea of assembling, we start with a simple case: assembling independent sub-models in Section 2. The key finding is that, as long as the assembly model is not dominated by one or a few sub-models, the quadratic term in the Taylor expansion Eq.(1) becomes small in $O(1)$-neighborhoods of the parameter space when the number of sub-models $m$ is tends to infinity. This means that as $m$ increases to infinity, the assembly model becomes a linear function of parameters. Since we put almost no requirements on the form of sub-models, it is the assembling process that results in the linearity of the assembly model. This case includes two-layer neural networks as examples.

For deep neural networks (Section 3), we consider a neural network as a multi-level assembly model each neuron is considered as an assembly model constructed iteratively from all the neurons from the previous layer. We then follow an inductive argument: using near-linearity of previous-layer pre-activation neurons to show the linearity of next-layer pre-activation neurons. Our key finding is that, when the network width is large, the neurons within the same layer become essentially independent to each other in the sense that their gradient directions are orthogonal in the parameter space. This orthogonality allows for the existence of a new coordinate system such that these neuron gradients are parallel to the new coordinate axes. Within the new coordinate system, one can apply the argument of assembling independent sub-models, and obtain the $O(1)$-neighborhood linearity of the pre-activation neurons, as well as the output of the neural network.

We further point out that this assembling viewpoint and the $O(1)$-neighborhood linearity can be extended to more general neural network architectures, e.g., DenseNet.

We end this section by commenting on a few closely related concepts and point out some of the differences.

**Boosting.** Boosting (Schapire, 1990; Freund, 1995), which is a popular method that combines multiple "weak" models to produce a powerful ensembe model, has a similar form with the assembly model, Eq.(4), see, e.g., Friedman et al. (2000). However, we note a few key differences between the two. First, in boosting, each "weak" model is trained *separately* on the dataset and the coefficients (i.e., $v_i$ in Eq.(4)) of the weak models are determined by the model performance. In training an assembly model, the "weak" sub-models (i.e., hidden layer neurons), are never directly evaluated on the training dataset, and the coefficients $v_i$ are considered as parameters of the assembly model and trained by gradient descent. Second, in boosting, different data samples may have different sample weights. In assembling, the data samples always have a uniform weight.

**Bagging and Reservoir computing.** Bagging (Breiman, 1996) and Reservoir computing (Jaeger, 2001) are two other ways of combining multiple sub-models to build a single model. In bagging, each sub-model is individually trained and the ensemble model is simply the average or the max of the sub-models. In reservoir computing, each sub-model is fixed, and only the coefficients of the linear combination are trainable.

**Notation.** We use bold lowercase letters, e.g., $\mathbf{v}$, to denote vectors, capital letters, e.g., $W$, to denote matrices. We use $\| \cdot \|$ to denote the Euclidean norm of vectors and spectral norm (i.e., matrix 2-norm) for matrices. We denote the set $\{1, 2, \cdots, n\}$ as $[n]$. We use $\nabla_{\mathbf{w}} f$ to denote the

partial derivative of $f$ with respect to $\mathbf{w}$. When $\mathbf{w}$ represents all of the parameters of $f$, we omit the subscript, i.e., $\nabla f$.

## 2 ASSEMBLING INDEPENDENT MODELS

In this section, we consider assembling a sufficiently large number $m$ of independent sub-models and show that the resulting assembly model is (approximately) linear in $O(1)$-neighborhood of the model parameter space.

**Ingredients: sub-models.** Let's consider a set of $m$ (sub-)models $\{g_i\}_{i=1}^m$, where each model $g_i$, with a set of model parameters $\mathbf{w}_i \in \mathcal{D}_i \subset \mathcal{P}_i = \mathbb{R}^{p_i}$, takes an input $\mathbf{x} \in \mathcal{D}_\mathbf{x} \subset \mathbb{R}^d$ and outputs a scalar prediction $g_i(\mathbf{w}_i; \mathbf{x})$. Here $\mathcal{P}_i$ is the parameter space of sub-model $g_i$ and $\mathcal{D}_\mathbf{x}$ is the domain of the input. By independence, we mean that any two distinct models $g_i$ and $g_j$ share no common model parameters:

$$\mathcal{P}_i \cap \mathcal{P}_j = \{0\}, \quad \forall i \neq j \in [m]. \tag{2}$$

In this sense, the change of $g_i$'s output, as a result of change of it parameters $\mathbf{w}_i$, does not affect the outputs of the other models $g_j$ where $j \neq i$.

In addition, we require that there is no dominating sub-models over the others. Specifically, we assume the following in this section:

**Assumption 1** (No dominating sub-models)**.** There exists a constant $c \in (0, 1)$ independent of $m$ such that, for any parameter setting $\{\mathbf{w}_i : \mathbf{w}_i \in \mathcal{D}_i\}_{i=1}^m$ and input $\mathbf{x} \in \mathcal{D}_\mathbf{x}$,

$$\frac{\text{median}[|g_1(\mathbf{w}_1; \mathbf{x})|, \cdots, |g_m(\mathbf{w}_m; \mathbf{x})|]}{\max[|g_1(\mathbf{w}_1; \mathbf{x})|, \cdots, |g_m(\mathbf{w}_m; \mathbf{x})|]} \geq c. \tag{3}$$

Furthermore, we assume that $\max[|g_1(\mathbf{w}_1; \mathbf{x})|, \cdots, |g_m(\mathbf{w}_m; \mathbf{x})|] \in [a, b] \subset \mathbb{R}$ for some constants $a > 0$ and $b > 0$.

**Remark 1.** This assumption guarantees that most of the outputs of the sub-models are at the same order, typically $O(1)$. It makes sure that the resulting assembly model is not dominated by one or a minor portion of the sub-models. This is typically seen for the neurons of neural networks.

We further make the following technical assumption, which is common in the literature.

**Assumption 2** (Twice differentiablity and smoothness)**.** Each sub-model $g_i$ is twice differentiable with respect to the parameters $\mathbf{w}_i$, and is $\beta$-smooth in the parameter space: there exists a positive number $\beta$ such that, for any $i \in [m]$,

$$\|\nabla g_i(\mathbf{w}_i; \mathbf{x}) - \nabla g_i(\mathbf{w}_i'; \mathbf{x})\| \leq \beta \|\mathbf{w}_i - \mathbf{w}_i'\|.$$

This assumption makes sure that for each $g_i$, the gradient is well-defined and its Hessian has a bounded spectral norm.

**Assembly model.** Based on these sub-models $\{g_i\}_{i=1}^m$, we construct an assembly model (or super-model) $f$, using linear combination as follows:

$$f(\theta; \mathbf{x}) := \frac{1}{s(m)} \sum_{i=1}^m v_i g_i(\mathbf{w}_i; \mathbf{x}), \tag{4}$$

where $v_i$ is the weight of the sub-model $g_i$, and $1/s(m)$, as a function of $m$, is the scaling factor. Here, we denote $\theta$ as the set of parameters of the assembly model, $\theta := (\mathbf{w}_1, \ldots, \mathbf{w}_m)$. The total number of parameters that $f$ has is $p = \sum_{i=1}^m p_i$. Typical choices for the weights $v_i$ are setting $v_i = 1$ for all $i \in [m]$, or random i.i.d. drawing $v_i$ from some zero-mean probability distribution.

**Remark 2.** For the ease of analysis and simplicity of notation, we assumed that the outputs of assembly model $f$ and sub-models $g_i$ are scalars. It is not difficult to see that our analysis below also applies to the scenarios of finite dimensional model outputs.

**The scaling factor** $1/s(m)$**.** The presence of the scaling factor $1/s(m)$ is to keep the output of assembly model $f$ at the order $O(1)$ w.r.t. $m$. In general, we expect that $s(m)$ grows with $m$. In particular, when $v_i$ are chosen i.i.d. from a probability distribution with mean zero, e.g., $\{-1, 1\}$,

the sum in Eq.(4) is expected to be of the order $\sqrt{m}$, under the Assumption 1. In this case, the scaling factor $1/s(m) = O(1/\sqrt{m})$.

Now, we will show that the assembly model $f$ has a small quadratic term in its Taylor expansion, when the size of the set of sub-models i.e. $m$, is sufficiently large.

**Theorem 1.** *Consider the assembly model $f$ constructed in Eq.(4) with each $v_i$ either set to be $1$ or randomly drawn from $\{-1, 1\}$, and suppose Assumption 1 and 2 hold. Given a positive number $R > 0$ and a parameter setting $\theta_0 \in \mathbb{R}^p$, for any $\theta \in \mathbb{R}^p$ such that $\|\theta - \theta_0\| \leq R$, the absolute value of the quadratic term in Taylor expansion Eq.(1) is bounded by:*

$$\left| \frac{1}{2}(\theta - \theta_0)^T H(\xi)(\theta - \theta_0) \right| \leq \frac{\beta R^2}{2s(m)}. \tag{5}$$

*Proof.* In what follows, we don't explicitly write out the dependence on $\theta$ and $\mathbf{x}$ in the Hessian $H := \frac{\partial^2 f}{\partial \theta^2}$, for simplicity of notation. We further denote $H_{g_i} := \frac{\partial^2 g_i}{\partial \mathbf{w}_i^2}$ as the Hessian of $g_i$.

We decompose the assembly model Hessian $H$ as a linear combination of $H_{g_i}$. By the definition of the assembly model in Eq.(4), an arbitrary entry $H_{jk}$ of the Hessian can be written as: $H_{jk} = \frac{1}{s(m)} \sum_{i=1}^m v_i \frac{\partial^2 g_i}{\partial \theta_j \partial \theta_k}$. Here $\theta_j, \theta_k$ are two individual parameters of the assembly model $f$. Note that $\frac{\partial^2 g_i}{\partial \theta_j \partial \theta_k}$ is non-zero, only if both $\theta_j$ and $\theta_k$ are parameters of sub-model $g_i$, i.e., $\theta_j, \theta_k \in \mathcal{P}_i$. Hence, the assembly model Hessian can be decomposed as a linear combination of sub-model Hessians: $H = \frac{1}{s(m)} \sum_{i=1}^m v_i H_{g_i}$. Therefore, the quadratic term becomes

$$\frac{1}{s(m)} \sum_{i=1}^m v_i \left[ \frac{1}{2}(\theta - \theta_0)^T H_{g_i}(\xi)(\theta - \theta_0) \right] = \frac{1}{s(m)} \sum_{i=1}^m v_i \left[ \frac{1}{2}(\mathbf{w}_i - \mathbf{w}_{i,0})^T H_{g_i}(\xi)(\mathbf{w}_i - \mathbf{w}_{i,0}) \right]$$

and its absolute value is bounded by

$$\left| \frac{1}{2}(\theta - \theta_0)^T H(\xi)(\theta - \theta_0) \right| \leq \frac{1}{2s(m)} \sum_{i=1}^m |v_i| \cdot \|H_{g_i}\| \cdot \|\mathbf{w}_i - \mathbf{w}_{i,0}\|^2 \leq \frac{\beta}{2s(m)} \sum_{i=1}^m \|\mathbf{w}_i - \mathbf{w}_{i,0}\|^2$$

In the second inequality, we used that $|v_i| = 1$ and that $g_i$ is $\beta$-smooth. Because of the independence of the sub-models as seen in Eq.(2), the summation in the above equation becomes $\|\theta - \theta_0\|^2$, which is bounded by $R^2$, as stated in the theorem condition. Therefore, we conclude the theorem. $\square$

It is important to note that $\beta$ is a constant and $1/s(m) = O(1/\sqrt{m})$. Then we have the following corollary in the limiting case.

**Corollary 1.** *Consider the assembly model $f$ under the same setting as in Theorem 1. If the number of sub-models $m$ increases to infinity, then for all parameter $\theta_0$ and input $\mathbf{x}$, the quadratic term*

$$\left| \frac{1}{2}(\theta - \theta_0)^T H(\xi)(\theta - \theta_0) \right| \to 0, \tag{6}$$

*as long as $\theta$ is within an $O(1)$-neighborhood of $\theta_0$, i.e., $\|\theta - \theta_0\|^2 \leq R$ for some constant $R > 0$.*

**Example: Two-layer neural networks.** A good example of this kind of assembly model is the two-layer neural network. A two-layer neural network is mathematically defined as:

$$f(W, \mathbf{v}; \mathbf{x}) = \frac{1}{\sqrt{m}} \sum_{i=1}^m u_i \sigma(\mathbf{w}_i^T \mathbf{x}), \tag{7}$$

where $m$ is the number of hidden layer neurons, $\sigma(\cdot)$ is the activation function, $\mathbf{x} \in \mathbb{R}^d$ is the network input, and $W \in \mathbb{R}^{m \times d}$ and $\mathbf{u} \in \mathbb{R}^m$ are the parameters for the first and second layer, respectively. Here, we can view the $i$-th hidden neuron and all the parameters $\mathbf{w}_i$ and $u_i$ that connect to it as the $i$-th sub-model: $g_i = u_i \sigma(\mathbf{w}_i^T \mathbf{x})$. We see that these sub-models do not share parameters, and each sub-model has $d + 1$ parameters. In addition, the weights of the sub-models are all $1$. By Corollary 1, the two-layer neural network becomes a linear model in any $O(1)$-neighborhoods, as the network width $m$ increases to infinity. This is consistent with the previous observation that a two-layer neural network transitions to linearity Liu et al. (2020).

**Gaussian distributed weights** $v_i$. Another common way to set the weights $v_i$ of the sub-models is to independently draw each $v_i$ from $\mathcal{N}(0, 1)$. In this case, $v_i$ is unbounded, but with high probability the quadratic term is still $O(\log(m)/\sqrt{m})$. Please see the detailed analysis in Appendix A.

**Hierarchy of assembly models.** In principle, we can consider the assembly model $f$, together with multiple similar models independent to each other, as sub-models, and construct a higher-level assembly model. Repeating this procedures, we can have a hierarchy of assembly models. Our analysis above also applies to this case and each assembly model is also $O(1)$-neighborhood linear, when the number of its sub-models is sufficiently large. However, one drawback of this hierarchy is that the total number of parameters of the highest level assembly model increase exponentially with the number of levels. We will see shortly that wide neural networks, as a hierarchy of assembly models, allows overlapping between sub-models, but still keeping the $O(1)$-neighborhood linearity.

## 3 DEEP NEURAL NETWORKS AS ASSEMBLING MODELS

In this section, we view deep neural networks as multi-level assembly models and show how the $O(1)$-neighborhood linearity arises naturally as a consequence of assembling.

**Setup.** We start with the definition of multi-layer neural networks. A $L$-layer full-connected neural network is defined as follows:

$$\alpha^{(0)} = \mathbf{x},$$
$$\alpha^{(l)} = \sigma(\tilde{\alpha}^{(l)}), \quad \tilde{\alpha}^{(l)} = \frac{1}{\sqrt{m_{l-1}}} W^{(l)} \alpha^{(l-1)}, \ \forall l = 1, 2, \cdots, L, \tag{8}$$
$$f = \tilde{\alpha}^{(L)},$$

where $\sigma(\cdot)$ is the activation function and is applied entry-wise above. We assume $\sigma(\cdot)$ is twice differentiable to make sure that the network $f$ is twice differentiable and its Hessian is well-defined. We also assume that $\sigma(\cdot)$ is an injective function, which includes the commonly used *sigmoid*, *tanh*, *softplus*, etc. In the network, with $m_l$ being the width of $l$-th hidden layer, $\tilde{\alpha}^{(l)} \in \mathbb{R}^{m_l}$ (called *pre-activations*) and $\alpha^{(l)} \in \mathbb{R}^{m_l}$ (called *post-activations*) represent the vectors of the $l$-th hidden layer neurons before and after the activation function, respectively. Denote $\theta := (W^{(1)}, \ldots, W^{(L)})$, with $W^{(l)} \in \mathbb{R}^{m_{l-1} \times m_l}$, as all the parameters of the network, and $\theta^{(l)} := (W^{(1)}, \ldots, W^{(l)})$ as the parameters before layer $l$. We also denote $p$ and $p^{(l)}$ as the dimension of $\theta$ and $\theta^{(l)}$, respectively. Denote $\theta(\alpha_i^{(l)})$ as the set of the parameters that neuron $\alpha_i^{(l)}$ depends on.

This neural network is typically initialized following Gaussian random initialization: each parameter is independently drawn from normal distribution, i.e., $(W_0^{(l)})_{ij} \sim \mathcal{N}(0, 1)$. In this paper, we focus on the $O(1)$-neighborhood of the initialization $\theta_0$. As pointed out by Liu et al. (2020), the whole gradient descent trajectory is contained in a $O(1)$-neighborhood of the initialization $\theta_0$ (more precisely, a Euclidean ball $B(\theta_0, R)$ with finite $R$).

**Overview of the methodology.** As it is defined recursively in Eq.(8), we view the deep neural network as a multi-level assembly model. Specifically, a pre-activation neuron at a certain layer $l$ is considered as an assembly model constructed from the post-activation neurons at layer $l - 1$, at the same time its post-activation also serves as a sub-model for the neurons in layer $l + 1$. Hence, a pre-activation $\tilde{\alpha}_i^{(l)}$ is an $l$-level assembly model, and the post-activation $\alpha_i^{(l)}$ is a $(l+1)$-level sub-model. To prove the $O(1)$-neighborhood linearity of the neural network, we start from the linearity at the first layer, and then follow an inductive argument from layer to layer, up to the output. The main argument lies in the inductive steps, from layer $l$ to layer $l + 1$. Our key finding is that, in the infinite width limit, the neurons within the same layer become independent to each other, which allows us using the arguments in Section 2 to prove the $O(1)$-neighborhood linearity. Since the exact linearity happens in the infinite width limit, in the following analysis we take $m_1, m_2, \ldots, m_{L-1} \to \infty$ sequentially, the same setting as in Jacot et al. (2018). Note that for neural networks that finite but large network width, the linearity will be approximate; the larger the width, the closer to linear.

### 3.1 BASE CASE: FIRST AND SECOND HIDDEN LAYER.

The first hidden layer pre-activations are defined as: $\tilde{\alpha}^{(1)} = \frac{1}{\sqrt{m_0}} W^{(1)} \mathbf{x}$, where $m_0 = d$. It is obvious that each element of $\tilde{\alpha}^{(1)}$ is linear in its parameters. Each second layer pre-activation $\tilde{\alpha}_i^{(1)}, i \in [m_2]$ can be considered as a two-layer neural network with parameters $\{W^{(1)}, \mathbf{w}_i^{(2)}\}$, where $\mathbf{w}_i^{(2)}$ is the $i$-th row of $W^{(2)}$. As we have seen in Section 2, the sub-models of a two-layer neural network share no common parameters, and the assembly model, which is the network itself, is $O(1)$-neighborhood linear in the limit of $m_1 \to \infty$.

### 3.2 FROM LAYER $l$ TO LAYER $l + 1$.

First, we make the induction hypothesis at layer $l$.

**Assumption 3** (Induction hypothesis). Assume that all the $l$-th layer pre-activation neurons $\tilde{\alpha}^{(l)}$ are $O(1)$-neighborhood linear in the limit of $m_1, \ldots, m_{l-1} \to \infty$.

Under this assumption, we will first show two key properties about the sub-models, i.e., the $l$-th layer post-activation neurons $\alpha^{(l)}$: (1) level sets of these neurons are hyper-planes with co-dimension 1 in $O(1)$-neighborhoods; (2) normal vectors of these hyper-planes are orthogonal with probability one in the infinite width limit.

**Setup.** Note that $l$-th layer neurons only depend on the parameters $\theta^{(l)}$. Without ambiguity, denote the parameter space spanned by $\theta^{(l)}$ as $\mathcal{P}$. We can write $\theta^{(l)}$ as

$$\theta^{(l)} = \theta^{(l-1)} \cup \mathbf{w}_1^{(l)} \cup \ldots \cup \mathbf{w}_{m_l}^{(l)},$$

where $\mathbf{w}_i^{(l)}$ is the $i$-th row of the weight matrix $W^{(l)}$. One observation is that the set of parameters of neuron $\alpha_i^{(l)}$ is $\theta(\alpha_i^{(l)}) = \theta^{(l-1)} \cup \mathbf{w}_i^{(l)}$. Namely, $\theta^{(l-1)}$ are shared by all $l$-layer neurons and each parameter in $\mathbf{w}_i^{(l)}$ is privately owned by only one neuron $\alpha_i^{(l)}$. Hence, we can decompose the parameter space $\mathcal{P}$ as: $\mathcal{P} = \bigoplus_{i=0}^{m_l} \mathcal{P}_i$, where common space $\mathcal{P}_0$ is spanned by $\theta^{(l-1)}$ and the private spaces $\mathcal{P}_i, i \neq 0$, are spanned by $\mathbf{w}_i^{(l)}$. Define a set of projection operators $\{\pi_i\}_{i=0}^{m_l}$ such that, for any vector $\mathbf{z} \in \mathcal{P}$, $\pi_i(\mathbf{z}) \in \mathcal{P}_i$, and $\sum_{i=0}^{m_l} \pi_i(\mathbf{z}) = \mathbf{z}$.

**Property 1: Level sets of neurons are hyper-planes.** The first key observation is that each of the $l$-th layer post-activations $\alpha^{(l)}$ has linear level sets in the $O(1)$-neighborhoods.

**Theorem 2** (Linear level sets of neurons). *Assume the induction hypothesis Assumption 3 holds. Given a fix input $\mathbf{x}$, for each post-activation $\alpha_i^{(l)}$, $i \in [m_l]$, the level set*

$$\mathcal{S}_c := \{\theta^{(l)} : \alpha_i^{(l)}(\theta^{(l)}; \mathbf{x}) = c\} \cap \mathcal{N}(\theta_0), \tag{9}$$

*is linear or an empty set, for all $c \in \mathbb{R}$, where $\mathcal{N}(\theta_0)$ is an $O(1)$-neighborhood of $\theta_0$. Moreover, these level sets are parallel to each other: for any $c_1, c_2 \in \mathbb{R}$, if $\mathcal{S}_{c_1} \neq \emptyset$ and $\mathcal{S}_{c_2} \neq \emptyset$, then $\mathcal{S}_{c_1}$ and $\mathcal{S}_{c_2}$ are parallel to each other.*

**Remark 3.** With abuse of notation, we did not explicitly write out the dependence of the level set on the neuron $\alpha_i^{(l)}$, the input $\mathbf{x}$, and the network initialization $\theta_0$.

*Proof.* First, note that all the pre-activation neurons $\tilde{\alpha}^{(l)}$ are linear in $\mathcal{N}(\theta_0)$ in the limit of $m_1, \ldots, m_{l-1} \to \infty$, as assumed in the induction hypothesis. This linearity of these function guarantees that all the level sets of the pre-activations $\{\theta^{(l)} : \tilde{\alpha}_i^{(l)}(\theta^{(l)}; \mathbf{x}) = c\} \cap \mathcal{N}(\theta_0)$, for all $c \in \mathbb{R}$ and $i \in [m_l]$ are either linear or empty sets, and they are parallel to each other. Since the activation function $\sigma(\cdot)$ is element-wisely applied to the neurons, it does not change the shape and direction of the level sets. Specifically, if $\{\theta^{(l)} : \alpha_i^{(l)}(\theta^{(l)}; \mathbf{x}) = c\}$ is non-empty, then

$$\{\theta^{(l)} : \alpha_i^{(l)}(\theta^{(l)}; \mathbf{x}) = c\} = \{\theta^{(l)} : \tilde{\alpha}_i^{(l)}(\theta^{(l)}; \mathbf{x}) = \sigma^{-1}(c)\}.$$

Therefore, $\{\theta^{(l)} : \alpha_i^{(l)}(\theta; \mathbf{x}) = c\} \cap \mathcal{N}(\theta_0)$ is linear or an empty set, and the activation function $\sigma(\cdot)$ preserves the parallelism. $\square$

This means that, although each of the neurons $\alpha^{(l)}$ is constructed by a large number of neurons in previous layers containing tremendously many parameters and is transformed by non-linear functions, it actually has a very simple geometric structure. A geometric view of the post-activation neurons is illustrated in Figure 1.

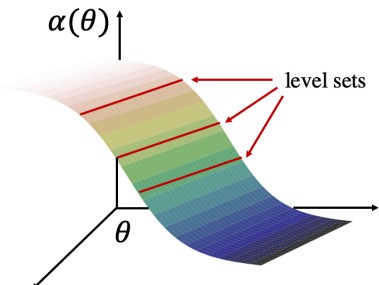

As the neurons have scalar outputs and $\sigma(\cdot)$ is injective by assumption, each level set is a piece of a co-dimension 1 hyper-plane in the $p$-dimensional parameter space $\mathcal{P}$. Hence, at each point of the hyper-plane there exists a unique (up to a negative sign) unit-length normal vector $\mathbf{n}$, which is perpendicular to the hyper-plane. A direct corollary of the linearity of the level sets, Theorem 2, is that the normal vector $\mathbf{n}$ is identical everywhere in the $O(1)$-neighborhood $\mathcal{N}(\theta_0)$.

Figure 1: A geometric view of the post-activation neurons. Level sets are parallel hyper-planes.

**Corollary 2.** *Assume the induction hypothesis Assumption 3 holds. Given a specific neuron $\alpha_i^{(l)}$ and an input $\mathbf{x}$, for any $\theta_1^{(l)}, \theta_2^{(l)} \in \mathcal{P} \cap \mathcal{N}(\theta_0)$, $\mathbf{n}(\theta_1^{(l)}) = \mathbf{n}(\theta_2^{(l)})$.*

Hence, we can define $\mathbf{n}_i$ as the normal vector for each neuron $\alpha_i^{(l)}$. The next property is about the set of normal vectors $\{\mathbf{n}_i\}_{i=1}^{m_l}$.

**Property 2: Orthogonality of normal vectors $\mathbf{n}_i$.** Now, let's look at the directions of the normal vectors $\mathbf{n}_i$. Note that the neuron $\alpha_i^{(l)}$ does not depend on the parameters $\mathbf{w}_j^{(l)}$ for any $j \neq i$, hence $\pi_j(\mathbf{n}_i) = \mathbf{0}$ for all $j \notin \{i, 0\}$. That means:

**Proposition 1.** *For all $i \in [m_l]$, the normal vector $\mathbf{n}_i$ resides in the sub-space $\mathcal{P}_i \oplus \mathcal{P}_0$, and can be decomposed as $\mathbf{n}_i = \pi_i(\mathbf{n}_i) + \pi_0(\mathbf{n}_i)$.*

As $\pi_i(\mathbf{n}_i) \in \mathcal{P}_i$, and $\mathcal{P}_i \cap \mathcal{P}_j = \{\mathbf{0}\}$ for $i \neq j$, the components $\{\pi_i(\mathbf{n}_i)\}_{i=1}^{m_l}$ are orthogonal to each other:

$$\pi_i(\mathbf{n}_i) \perp \pi_j(\mathbf{n}_j), \quad for\ all\ i \neq j \in [m_l]. \tag{10}$$

By Proposition 1, to show the orthogonality of the normal vectors $\{\mathbf{n}_i\}_{i=1}^{m_l}$, it suffices to show the orthogonality of $\{\pi_0(\mathbf{n}_i)\}_{i=1}^{m_l}$.

Since it is perpendicular to the corresponding level sets, the normal vector $\mathbf{n}_i$ is parallel to the gradient $\nabla \alpha_i^{(l)}$, up to a potential negative sign. Similarly, the projection $\pi_0(\mathbf{n}_i)$ is parallel to the partial gradient $\nabla_{\theta^{(l-1)}} \tilde{\alpha}_i^{(l)}$.

By the definition of neurons in Eq.(8) and the constancy of $\mathbf{n}_i$ in the neighborhood $\mathcal{N}(\theta_0)$, we have

$$(\nabla_{\theta^{(l-1)}} \tilde{\alpha}_i^{(l)})^T = \frac{1}{\sqrt{m_{l-1}}} (\mathbf{w}_{i,0}^{(l)})^T \nabla \alpha^{(l-1)}. \tag{11}$$

Here, because $\alpha^{(l-1)}$ is an $m_l$-dimensional vector, $\nabla \alpha^{(l-1)}$ is an $m_l \times p^{(l-1)}$ matrix, where $p^{(l-1)}$ denotes the size of $\theta^{(l-1)}$. It is important to note that the matrix $\nabla \alpha^{(l-1)}$ is shared by all the neurons in layer $l$ and is independent of the index $i$, while the vector $\mathbf{w}_{i,0}^{(l)}$, which is $\mathbf{w}_i^{(l)}$ at initialization, is totally private to the $l$-th layer neurons.

Recall that the vectors $\{\mathbf{w}_{i,0}^{(l)}\}_{i=1}^{m_l}$ are independently drawn from $\mathcal{N}(\mathbf{0}, I_{m_{l-1} \times m_{l-1}})$. As is well-known, when the vector dimension $m_{l-1}$ is large, these independent random vectors $\{\mathbf{w}_{i,0}^{(l)}\}$ are nearly orthogonal. When in the infinite width limit, the orthogonality holds with probability 1:

$$\forall \epsilon > 0, \quad \lim_{m_{l-1} \to \infty} \mathbb{P} \left( \frac{1}{m_{l-1}} |(\mathbf{w}_{i,0}^{(l)})^T \mathbf{w}_{j,0}^{(l)}| \geq \epsilon \right) = 0, \quad for\ all\ i \neq j \in [m_l]. \tag{12}$$

From Eq.(11), we see that the partial gradients $\{\nabla_{\theta^{(l-1)}} \tilde{\alpha}_i^{(l)}\}_{i=1}^{m_l}$ are in fact the result of applying a universal linear transform, i.e., $\nabla \alpha^{(l-1)}$, onto a set of nearly orthogonal vectors. The following lemma shows that the vectors remain orthogonal even after this linear transformation.

**Lemma 1.**

$$\forall \epsilon > 0, \quad \lim_{m_{l-1} \to \infty} \mathbb{P}\left(\left|(\nabla_{\theta^{(l-1)}} \tilde{\alpha}_i^{(l)})^T \nabla_{\theta^{(l-1)}} \tilde{\alpha}_j^{(l)}\right| \geq \epsilon\right) = 0, \quad for \ all \ i \neq j \in [m_l]. \tag{13}$$

See the proof in Appendix C.

Recalling Eq.(10) and the fact that the normal vectors $\pi_0(\mathbf{n}_i)$ are parallel to the gradients $\nabla_{\theta^{(l-1)}} \tilde{\alpha}^{(l)}$, we immediately have that these normal vectors are orthogonal with probability 1, in the limit of $m_{l-1} \to \infty$, as stated in the following theorem.

**Theorem 3** (Orthogonality of normal vectors).

$$\forall \epsilon > 0, \quad \lim_{m_{l-1} \to \infty} \mathbb{P}\left(\left|\mathbf{n}_i^T \mathbf{n}_j\right| \geq \epsilon\right) = 0, \quad for \ all \ i \neq j \in [m_l]. \tag{14}$$

**Remark 4.** As seen in Section 2, a two-layer neural network Eq.(7) is an assembly model with independent sub-models. The normal vectors $\{\mathbf{n}_i\}_{i=1}^{m_l}$ are exactly orthogonal to each other, even for finite hidden layer width.

See Appendix B for an numerical verification of this orthogonality. This orthogonality allows the existence of a new coordinate system such that all the normal vectors are along the axes. Specifically, in the old coordinate system $\mathcal{O}$, each axis is along an individual neural network parameter $\theta_i \in \theta$ with $i \in [p]$. After an appropriate rotation, $\mathcal{O}$ can be transformed to a new coordinate system $\mathcal{O}'$ such that each normal vector $n_i$ is parallel to one of the new axes. See Figure 2 for an illustration. Denote $\theta'$ as the set of new parameters that are long the axes of new coordinate system $\mathcal{O}'$: $\theta' = \mathcal{R}\theta$, where $\mathcal{R}$ is a rotation operator.

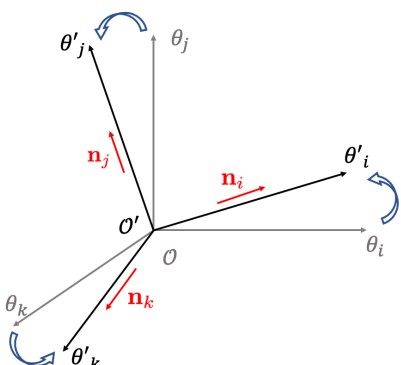

Figure 2: Coordinate systems: $\mathcal{O}'$ (new) vs. $\mathcal{O}$ (old). normal vectors $\mathbf{n}_i$ and gradients $\nabla \alpha_i$ are along an axis of $\mathcal{O}'$.

The interesting observation is that each neuron $\alpha_i^{(l)}$ essentially depends on only one new parameter $\theta_i' \in \theta'$, because its gradient direction is parallel with one normal vector $\mathbf{n}_i$ and $\mathbf{n}_i$ is along one axis in the new coordinate system $\mathcal{O}'$. Moreover, different neurons depends on different new parameters, as normal vectors are never parallel. Hence, these neurons are essentially independent to each other. This view actually allows us to use the analysis for assembling independent models in Section 2 to show the $O(1)$-neighborhood linearity at layer $l + 1$, as follows.

**Linearity of the pre-activations in layer** $l+1$**.** Having the properties for neurons in layer $l$ discussed above, we are now ready to analyze the pre-activation neurons $\tilde{\alpha}^{(l+1)}$ in layer $l + 1$ as assembly models. Recall that each pre-activation $\tilde{\alpha}^{(l+1)}$ is defined as:

$$\tilde{\alpha}^{(l+1)} = \frac{1}{\sqrt{m_l}} \sum_{i=1}^{m_l} w_j^{(l+1)} \alpha_j^{(l)}. \tag{15}$$

Without ambiguity, we omitted the index $i$ for the pre-activation $\tilde{\alpha}^{(l+1)}$ and the weights $\mathbf{w}^{(l+1)}$.

We want to derive the quadratic term (i.e., the Lagrange remainder term) of $\tilde{\alpha}^{(l+1)}$ in its Taylor expansion and to show it is arbitrarily small, in the $O(1)$-neighborhood $\mathcal{N}(\theta_0)$. First, let's consider the special case where the parameters $\mathbf{w}^{(l+1)}$ are fixed, i.e., $\mathbf{w}^{(l+1)} = \mathbf{w}_0^{(l+1)}$, and $\tilde{\alpha}^{(l+1)}$ only depends on $\theta^{(l)}$. This case conveys the key concepts of the $O(1)$-neighborhood linearity after assembling. We will relax this constraint in Appendix D.

Consider an arbitrary parameter setting $\theta \in \mathcal{N}(\theta_0)$, and let $R := \|\theta - \theta_0\| = O(1)$. By the definition of assembly model $\tilde{\alpha}^{(l+1)}$ in Eq.(15), the quadratic term in its Taylor expansion Eq.(1) can be decomposed as:

$$\frac{1}{2}(\theta - \theta_0)^T H_{\tilde{\alpha}^{(l+1)}}(\xi)(\theta - \theta_0) = \frac{1}{\sqrt{m_l}} \sum_{i=1}^{m_l} w_{i,0}^{(l+1)} \left[\frac{1}{2}(\theta - \theta_0)^T H_{\alpha_i^{(l)}}(\xi)(\theta - \theta_0)\right]. \tag{16}$$

where $H_{\tilde{\alpha}^{(l+1)}} = \frac{\partial^2 \tilde{\alpha}^{(l+1)}}{\partial \theta^2}$ and $H_{\alpha_i^{(l)}} = \frac{\partial^2 \alpha_i^{(l)}}{\partial \theta^2}$ are the Hessians of $\tilde{\alpha}^{(l+1)}$ and $\alpha_i^{(l)}$, respectively, and $\xi \in \mathcal{P}$ is on the line segment between $\theta^{(l)}$ and $\theta_0^{(l)}$.

We bounded the term in the square bracket using a treatment analogous to Theorem 1. First, as seen in Property 1, the level sets of $\alpha_i^{(l)}$ are hyper-planes with co-dimension 1, perpendicular to $\mathbf{n}_i$. Then the value of $\alpha_i^{(l)}$ only depends on the component $(\theta - \theta_0)^T \mathbf{n}_i$, and Hessian $H_{\alpha_i^{(l)}}$ is rank 1 and can be written as $H_{\alpha_i^{(l)}} = c \mathbf{n}_i \mathbf{n}_i^T$, with some constant $c \leq \beta$. Hence, we have

$$\left| (\theta - \theta_0)^T H_{\alpha_i^{(l)}}(\xi)(\theta - \theta_0) \right| = c \left( (\theta - \theta_0)^T \mathbf{n}_i \right)^2 \leq \beta \left( (\theta - \theta_0)^T \mathbf{n}_i \right)^2 .$$

Here $\beta$ is the smoothness of the sub-model, i.e., post-activation $\alpha^{(l)}$. By Eq.(16), we have:

$$\frac{1}{2} \left| (\theta - \theta_0)^T H_{\tilde{\alpha}^{(l+1)}}(\xi)(\theta - \theta_0) \right| \leq \frac{\beta}{2\sqrt{m_l}} \max(|w_{i,0}^{(l+1)}|) \sum_{i=1}^{m_l} \left( (\theta - \theta_0)^T \mathbf{n}_i \right)^2 . \qquad (17)$$

Second, by the orthogonality of normal vectors as in Theorem 3, we have

$$\sum_{i=1}^{m_l} \left( (\theta - \theta_0)^T \mathbf{n}_i \right)^2 \leq \|\theta - \theta_0\|^2 = R^2 . \qquad (18)$$

Combining the above two equations, we obtain the following theorem which upper bound the magnitude of the quadratic term of $\tilde{\alpha}^{(l+1)}$:

**Theorem 4** (Bounding the quadratic term). *Assume that the parameters in layer $l+1$ are fixed to the initialization. For any parameter setting $\theta \in \mathcal{N}(\theta_0)$,*

$$\frac{1}{2} \left| (\theta - \theta_0)^T H_{\tilde{\alpha}^{(l+1)}}(\xi)(\theta - \theta_0) \right| \leq \frac{\beta R}{2\sqrt{m_l}} \max(|w_{i,0}^{(l+1)}|). \qquad (19)$$

For the Gaussian random initialization, the above upper bound is of the order $O(\log(m_l)/\sqrt{m_l})$. Hence, as $m_l \to \infty$, the quadratic term of pre-activation $\tilde{\alpha}^{(l+1)}$ in the $(l+1)$-th layer vanishes, and the function $\tilde{\alpha}^{(l+1)}$ becomes linear.

**Concluding the induction analysis.** Applying the analysis in Section 3.2, Theorem 2 - 4, with a standard induction argument, we conclude that the neural network, as well as each of its hidden pre-activation neurons, is $O(1)$-neighborhood linear, in the infinite network width limit, $m_1, \ldots, m_{L-1} \to \infty$.

**Extension to other architectures.** In Appendix E, we further show that the assembly analysis and the $O(1)$-neighborhood linearity also hold for DenseNet (Huang et al., 2017).

# 4 CONCLUSION AND FUTURE DIRECTIONS

In this work, we viewed a wide fully-connected neural network as a hierarchy of assembly models. Each assembly model corresponds to a pre-activation neuron and is linearly assembled from a set of sub-models, the post-activation neurons from the previous layer. When the network width increases to infinity, we observed that the neurons within the same hidden layer become essentially independent. With this property, we shown that the network is linear in an $O(1)$-neighborhood around the network initialization.

We believe the assembly analysis and the principles we identified, especially the essential independence of sub-models, and their iterative construction, are significantly more general than the specific structures we considered in this work, and, for example, apply to a broad range of neural architectures. In future work, we aim to apply the analysis to general feed-forward neural networks, such as architectures with arbitrary connections that form an acyclic graph.

## ACKNOWLEDGEMENTS

We are grateful for the support of the NSF and the Simons Foundation for the Collaboration on the Theoretical Foundations of Deep Learning[1] through awards DMS-2031883 and #814639. We also acknowledge NSF support through IIS-1815697 and the TILOS institute (NSF CCF-2112665).

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

## A    RANDOM SUB-MODEL WEIGHTS WITH GAUSSIAN DISTRIBUTION

Let's consider the case where the weights $v_i$ of the sub-models are randomly drawing from Gaussian distributions, for example the normal distribution $\mathcal{N}(0,1)$. That is

$$f(\theta; \mathbf{x}) = \frac{1}{s(m)} \sum_{i=1}^{m} v_i g_i(\mathbf{w}_i; \mathbf{x}), \tag{20}$$

where $v_i$'s are randomly drawn from $\mathcal{N}(0,1)$.

In this case, there is no strict upper bounded on the absolute values of weights $|v_i|$. But with high probability, we can still bound them using the following lemma.

**Lemma 2.** *Let $v_1, ..., v_m$ be i.i.d. Gaussian variables. Then with probability at least $1 - 2e^{-0.5 \log^2 m + \log m}$,*

$$\max_{i \in [m]} |v_i| \le \log m.$$

The above lemma can be obtained by letting $t = \log m$ in Eq.(2.10) in Vershynin (2018) and using union bound.

Using the same analysis as in the proof of Theorem 1, we have, with probability at least $1 - 2e^{-0.5 \log^2 m + \log m}$,

$$\left| \frac{1}{2} (\theta - \theta_0)^T H(\xi)(\theta - \theta_0) \right| \le \frac{1}{2s(m)} \sum_{i=1}^{m} |v_i| \cdot \|H_{g_i}\| \cdot \|\mathbf{w}_i - \mathbf{w}_{i,0}\|^2 \le \frac{\beta \log m}{2s(m)} \sum_{i=1}^{m} \|\mathbf{w}_i - \mathbf{w}_{i,0}\|^2$$

Under Assumption 2, $\beta$ is a constant. Note that as the number of $m$ goes to infinity, the probability $1 - 2e^{-0.5 \log^2 m + \log m}$ converges to 1. Since $1/s(m)$ is $O(1/\sqrt{m})$, hence, we get

$$\left| \frac{1}{2} (\theta - \theta_0)^T H(\xi)(\theta - \theta_0) \right| = O\left( \frac{\log m}{\sqrt{m}} \right). \tag{21}$$

When $m \to \infty$, the quadratic term of the Taylor expansion converges to 0, with probability 1.

## B    EXPERIMENTAL VERIFICATION OF THE ORTHOGONALITY IN THEOREM 3

In this section, we run experiments to verify the theoretical finding in Theorem 3, that the normal vectors $\mathbf{n}_i$ within the same hidden layer tends to become more and more orthogonal, as the layer width increases.

Specifically, we run two experiments. In the first one, we use full-batch gradient descent to train 4-layer fully-connected neural networks on a simple dataset $\mathcal{D}$: 20 images randomly selected from CIFAR-10 of the classes "airplane" or "bird". In the network, all the hidden layers have the same width $m$. Given a training sample and any two neurons $\tilde{\alpha}_i$ and $\tilde{\alpha}_j$ in the same layer, we compute the cosine $\cos \theta_{ij}$ between the gradients $\nabla \tilde{\alpha}_i$ and $\nabla \tilde{\alpha}_j$ as

$$\cos \theta_{ij} = \frac{\langle \nabla \tilde{\alpha}_i, \nabla \tilde{\alpha}_j \rangle}{|\nabla \tilde{\alpha}_i| \cdot |\nabla \tilde{\alpha}_i|}. \tag{22}$$

As a metric to measure the orthogonality of the gradients, the average absolute cosine $|\cos \theta|$ is calculated by averaging $|\cos \theta_{ij}|$ over all pairs of $(i,j)$ with $i \ne j$ and over all data samples. Left panel of Figure 3 shows the numerical results of the average absolute cosine $|\cos \theta|$ as a function of network width $m$. We see that, as $m$ increases, $|\cos \theta|$ monotonically decreases towards zero, verifying that the neuron gradients becomes more and more orthogonal.

In the second experiment, we consider a bottleneck network, which has three hidden layers, with the 1st and 3rd layers with large width $m = 1000$ and 2nd layer with width $m_b$ as the bottleneck layer. We train this bottleneck network using full-batch gradient descent on the same dataset $\mathcal{D}$ as in the first experiment. Given a training sample and any two neurons $\tilde{\alpha}_i$ and $\tilde{\alpha}_j$ in the bottleneck layer (i.e., 2nd hidden layer), we compute the cosine of the angle between their gradients using Eq. (22),

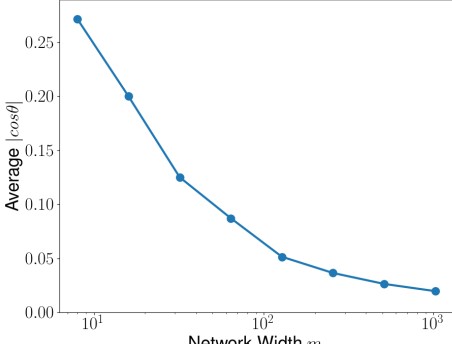 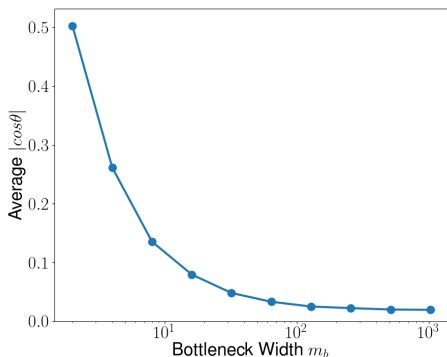

Figure 3: Absolute cosine of the angle between two neurons, averaged over all neuron pairs and data samples. **Left:** 4-layer fully-connected neural network, all hidden layers of which have equal width $m$; **Right:** Bottleneck network with three hidden layers, $m_1 = m_3 = 1000$ and $m_2 = m_b$. In both cases, the average absolute cosine $|\cos \theta|$ monotonically decreases towards zero, consistent with the theoretical prediction of orthogonality between neuron gradient in Theorem 3.

and then take the average of its absolute value over all pairs of different neurons and over all data samples. We use this averaged absolute cosine $|\cos \theta|$ as the metric to measure the orthogonality of the gradients. Right panel of Figure 3 shows the relation between $|\cos \theta|$ and the bottleneck width $m_b$. It is clear that $|\cos \theta|$ monotonically decreases towards zero as $m_b$ increase, verifying the orthogonality of gradients of bottleneck neurons.

## C PROOF OF LEMMA 1

As in Eq. (11), the partial gradient $\nabla_{\theta^{(l-1)}} \tilde{\alpha}_i^{(l)}$ can be written as: $\left(\nabla_{\theta^{(l-1)}} \tilde{\alpha}_i^{(l)}\right)^T = \frac{1}{\sqrt{m_{l-1}}} (\mathbf{w}_{i,0}^{(l)})^T \nabla \alpha^{(l-1)}$. Hence, we have

$$\left(\nabla_{\theta^{(l-1)}} \tilde{\alpha}_i^{(l)}\right)^T \nabla_{\theta^{(l-1)}} \tilde{\alpha}_j^{(l)} = \frac{1}{m_{l-1}} (\mathbf{w}_{i,0}^{(l)})^T \nabla \alpha^{(l-1)} (\nabla \alpha^{(l-1)})^T (\mathbf{w}_{j,0}^{(l)}).$$

Note that $\mathbf{w}_{i,0}^{(l)}$ and $\mathbf{w}_{j,0}^{(l)}$ are independent to each other when $i \neq j$, and $\nabla \alpha^{(l-1)}$ is a fixed matrix independent of $\mathbf{w}_{i,0}^{(l)}$ and $\mathbf{w}_{j,0}^{(l)}$. Denote $\mathbf{a}_j := \nabla \alpha^{(l-1)} (\nabla \alpha^{(l-1)})^T \mathbf{w}_{j,0}^{(l)} \in \mathbb{R}^{m_{l-1}}$. Then,

$$\left(\nabla_{\theta^{(l-1)}} \tilde{\alpha}_i^{(l)}\right)^T \nabla_{\theta^{(l-1)}} \tilde{\alpha}_j^{(l)} = \frac{1}{m_{l-1}} (\mathbf{w}_{i,0}^{(l)})^T \mathbf{a}_j, \tag{23}$$

the inner product of two independent vectors $\mathbf{w}_{i,0}^{(l)}$ and $\mathbf{a}_j$ with a scaling factor $1/m_{l-1}$.

When conditioned on the vector $\mathbf{a}_j$, the quantity in Eq.(23) is a Gaussian variable:

$$\frac{1}{m_{l-1}} (\mathbf{w}_{i,0}^{(l)})^T \mathbf{a}_j \sim \mathcal{N} \left( 0, \frac{\mathbf{a}_j^T \mathbf{a}_j}{m_{l-1}^2} \right),$$

because $\mathbf{w}_i^{(l)}$ is initialized following $\mathcal{N}(\mathbf{0}, I)$. In the limit of $m_{l-1} \to \infty$, if the variance $\frac{\mathbf{a}_j^T \mathbf{a}_j}{m_{l-1}^2}$ converges to 0, then this Gaussian variable should also converges to zero with probability 1, from which we can conclude the lemma. In the following, we will show that the variance $\frac{\mathbf{a}_j^T \mathbf{a}_j}{m_{l-1}^2}$ converges to 0 with probability 1 in this limit.

First, we need the following lemma to upper bound the spectral norm of $\nabla \alpha^{(l)}$. The proof is deferred to Section C.1.

**Lemma 3.** *Consider the neural networks defined in Eq.(8) with random Gaussian initialized parameters. There exist a constant $C > 0$, such that, with probability at least $1 - 4(l+1)e^{-m/2}$, we have*

$$\|\nabla\alpha^{(l)}(\nabla\alpha^{(l)})^T\| \le C,$$

*with $m = \min\{m_1, m_2, \ldots, m_{l-1}\}$.*

Since $\mathbf{a}_j = \nabla\alpha^{(l-1)}(\nabla\alpha^{(l-1)})^T\mathbf{w}_{j,0}^{(l)}$, using Lemma 3, we can bound $\|\mathbf{a}_j^T\mathbf{a}_j\|$ by

$$\|\mathbf{a}_j^T\mathbf{a}_j\| \le \|\nabla\alpha^{(l-1)}(\nabla\alpha^{(l-1)})^T\|^2\|\mathbf{w}_{j,0}^{(l)}\|^2 \le C^2\|\mathbf{w}_{j,0}^{(l)}\|^2.$$

Recall that $\mathbf{w}_{j,0}^{(l)} \in \mathbb{R}^{m_{l-1}}$ follows the Gaussian distribution $\mathcal{N}(\mathbf{0}, I)$. Then $\|\mathbf{w}_{j,0}^{(l)}\|^2 \sim \chi^2(m_{l-1})$. By Lemma 1 in Laurent & Massart (2000), we have with probability at least $1 - e^{-m_{l-1}}$,

$$\|\mathbf{w}_{j,0}^{(l)}\|^2 \le 5m_{l-1}.$$

By union bound, we have with probability $1 - 2(L+1)e^{-m/2} - e^{-m_{l-1}}$, the variance

$$\frac{\mathbf{a}_j^T\mathbf{a}_j}{m_{l-1}^2} \le \frac{5C^2}{m_{l-1}}$$

In the infinite network width limit $m_1, \ldots, m_{l-1} \to \infty$, we see that the variance converges to $0$ and the probability converges to $1$.

### C.1 PROOF OF LEMMA 3

*Proof.* First, note that $\nabla\alpha^{(l-1)}(\nabla\alpha^{(l-1)})^T$ can be decomposed as

$$\nabla\alpha^{(l-1)}(\nabla\alpha^{(l-1)})^T = \sum_{l'=1}^{l-1} \nabla_{W^{(l')}}\alpha^{(l-1)}(\nabla_{W^{(l')}}\alpha^{(l-1)})^T.$$

Then, its spectral norm can be bounded by

$$\|\nabla\alpha^{(l-1)}(\nabla\alpha^{(l-1)})^T\| = \|\sum_{l'=1}^{l-1} \nabla_{W^{(l')}}\alpha^{(l-1)}(\nabla_{W^{(l')}}\alpha^{(l-1)})^T\|$$

$$\le \sum_{l'=1}^{l-1} \|\nabla_{W^{(l')}}\alpha^{(l-1)}(\nabla_{W^{(l')}}\alpha^{(l-1)})^T\|$$

$$\le \sum_{l'=1}^{l-1} \left\| \frac{\partial\alpha^{(l')}}{\partial W^{(l')}} \prod_{l''=l'+1}^{l-1} \frac{\partial\alpha^{(l'')}}{\partial\alpha^{(l''-1)}} \left( \frac{\partial\alpha^{(l')}}{\partial W^{(l')}} \prod_{l''=l'+1}^{l-1} \frac{\partial\alpha^{(l'')}}{\partial\alpha^{(l''-1)}} \right)^T \right\|$$

$$\le \sum_{l'=1}^{l-1} \left\| \frac{\partial\alpha^{(l')}}{\partial W^{(l')}} \right\|^2 \prod_{l''=l'+1}^{l-1} \left\| \frac{\partial\alpha^{(l'')}}{\partial\alpha^{(l''-1)}} \right\|^2. \tag{24}$$

In the following, we need to bound the terms $\|\partial\alpha^{(l')}/\partial W^{(l')}\|$ and $\|\partial\alpha^{(l'')}/\partial\alpha^{(l''-1)}\|$.

To simplify the presentation of the proof, we assume in the following that each network hidden layer has the same width $m$. We leave the general analysis for the readers.

We use the following lemma to bound the spectral norm of the weight matrices at initialization.

**Lemma 4.** *If each component of $W_0^{(l)}$, $l \in [L]$, is i.i.d. drawn from $\mathcal{N}(0, 1)$, then with probability at least $1 - 2e^{-m/2}$,*

$$\|W_0^{(l)}\| \le 3\sqrt{m}.$$

See the proof in Section C.2.

There is also a lemma that bounds the Euclidean norm of each hidden layer neuron vector $\alpha^{(l)}$.

**Lemma 5** (Modified Lemma F.3 of (Liu et al., 2020)). *There exists constants $C_\alpha, B_\alpha > 0$ such that, with probability at least $1 - 2(L+1)e^{-m/2}$, for all $l \in [L]$,*

$$\|\alpha^{(l)}\| \leq C_\alpha \sqrt{m} + B_\alpha. \tag{25}$$

Using the above two lemmas, we can bounded those two terms. At initialization, we have, with probability at least $1 - 2e^{-m/2}$,

$$\left\|\frac{\partial \alpha^{(l)}}{\partial \alpha^{(l-1)}}\right\|^2 = \sup_{\|\mathbf{v}\|=1} \frac{1}{m} \sum_{i=1}^{m} \left(\sigma'(\tilde{\alpha}_i^{(l)})W_{0,ij}^{(l)}v_j\right)^2$$

$$= \sup_{\|\mathbf{v}\|=1} \frac{1}{m}\|\Sigma'^{(l)}W_0^{(l)}\mathbf{v}\|^2$$

$$\leq \frac{1}{m}\|\Sigma'^{(l)}\|^2\|W_0^{(l)}\|^2$$

$$\leq 9\mathsf{L}_\sigma^2,$$

and with probability at least $1 - 2(L+1)e^{-m/2}$, for all $l \in [L]$,

$$\left\|\frac{\partial \alpha^{(l)}}{\partial W^{(l)}}\right\|^2 = \sup_{\|V\|_F=1} \frac{1}{m} \sum_{i=1}^{m} \left(\sum_{j,j'} \sigma'(\tilde{\alpha}_i^{(l)})\alpha_{j'}^{(l-1)}\mathbb{I}_{i=j}V_{jj'}\right)^2$$

$$= \sup_{\|V\|_F=1} \frac{1}{m}\|\Sigma'^{(l)}V\alpha^{(l-1)}\|^2$$

$$\leq \frac{1}{m}\|\Sigma'^{(l)}\|^2\|\alpha^{(l-1)}\|^2$$

$$\leq \mathsf{L}_\sigma^2 C_\alpha^2 + \frac{1}{m}\mathsf{L}_\sigma^2 B_\alpha^2. \tag{26}$$

Here $\Sigma'^{(l)}$ is a diagonal matrix, with the diagonal entry $\Sigma'^{(l)}_{ii} = \sigma'(\tilde{\alpha}_i^{(l)})$ and $\mathsf{L}_\sigma$ is the degree of Lipschitz continuity of the activation function $\sigma(\cdot)$.

Now, using the above results, we can upper bound the spectral norm of the matrix $\nabla\alpha^{(l-1)}(\nabla\alpha^{(l-1)})^T$. With probability $1 - 4(L+1)e^{-m/2}$, we have

$$\|\nabla\alpha^{(l-1)}(\nabla\alpha^{(l-1)})^T\| \leq \sum_{l'=1}^{l-1} \left\|\frac{\partial \alpha^{(l')}}{\partial W^{(l')}}\right\|^2 \prod_{l''=l'+1}^{l-1} \left\|\frac{\partial \alpha^{(l'')}}{\partial \alpha^{(l''-1)}}\right\|^2$$

$$\leq (l-1)9^{l-1}\mathsf{L}_\sigma^{2l}\left(C_\alpha^2 + \frac{1}{m}B_\alpha^2\right). \tag{27}$$

Since $l \leq L$, we can see that, for sufficient large network width $m$, the spectral norm $\|\nabla\alpha^{(l-1)}(\nabla\alpha^{(l-1)})^T\|$ is bounded by a constant. $\qquad\square$

### C.2 PROOF OF LEMMA 4

*Proof.* Consider an arbitrary random matrix $A \in \mathbb{R}^{m_a \times m_b}$ with each entry $A_{i,j} \sim \mathcal{N}(0,1)$. By Corollary 5.35 of Vershynin (2010), for any $t > 0$, we have with probability at least $1 - 2\exp(-\frac{t^2}{2})$,

$$\|A\| \leq \sqrt{m_a} + \sqrt{m_b} + t. \tag{28}$$

For the initial neural network weight matrices, we have

$$\|W_0^{(1)}\|_2 \leq \sqrt{d} + \sqrt{m} + t,$$

$$\|W_0^{(l)}\|_2 \leq 2\sqrt{m} + t, \quad l \in \{2, 3, ..., L\},$$

$$\|W_0^{(L+1)}\|_2 \leq \sqrt{m} + 1 + t.$$

Letting $t = \sqrt{m}$ and noting that $m > d$, we finish the proof.

$\qquad\square$

# D   INCLUDING $\mathbf{w}^{(l+1)}$ AS PARAMETERS

Now, we consider the more general case where parameters in layer $l + 1$ are not fixed. Then, the quadratic term of the pre-activation $\tilde{\alpha}^{(l+1)}$ is written as:

$$\frac{1}{2}(\theta - \theta_0)^T H_{\tilde{\alpha}^{(l+1)}}(\xi)(\theta - \theta_0)$$

$$= \frac{1}{2}\begin{pmatrix} \mathbf{w}^{(l+1)} - \mathbf{w}_0^{(l+1)} \\ \theta^{(l)} - \theta_0^{(l)} \end{pmatrix}^T \begin{bmatrix} \frac{\partial^2 \tilde{\alpha}^{(l+1)}}{(\partial \mathbf{w}^{(l+1)})^2}(\xi) & \frac{\partial^2 \tilde{\alpha}^{(l+1)}}{\partial \theta^{(l)} \partial \mathbf{w}^{(l+1)}}(\xi) \\ \frac{\partial^2 \tilde{\alpha}^{(l+1)}}{\partial \mathbf{w}^{(l+1)} \partial \theta^{(l)}}(\xi) & \frac{\partial^2 \tilde{\alpha}^{(l+1)}}{(\partial \theta^{(l)})^2}(\xi) \end{bmatrix} \begin{pmatrix} \mathbf{w}^{(l+1)} - \mathbf{w}_0^{(l+1)} \\ \theta^{(l)} - \theta_0^{(l)} \end{pmatrix}$$

$$= \frac{1}{2}(\mathbf{w}^{(l+1)} - \mathbf{w}_0^{(l+1)})^T \frac{\partial^2 \tilde{\alpha}^{(l+1)}}{(\partial \mathbf{w}^{(l+1)})^2}(\xi)(\mathbf{w}^{(l+1)} - \mathbf{w}_0^{(l+1)}) \quad (=: \mathcal{A})$$

$$+ \frac{1}{2}(\theta^{(l)} - \theta_0^{(l)})^T \frac{\partial^2 \tilde{\alpha}^{(l+1)}}{(\partial \theta^{(l)})^2}(\xi)(\theta^{(l)} - \theta_0^{(l)}) \quad (=: \mathcal{B})$$

$$+ (\mathbf{w}^{(l+1)} - \mathbf{w}_0^{(l+1)})^T \frac{\partial^2 \tilde{\alpha}^{(l+1)}}{\partial \mathbf{w}^{(l+1)} \partial \theta^{(l)}}(\xi)(\theta^{(l)} - \theta_0^{(l)}) \quad (=: \mathcal{C})$$

where $\xi$ is some point between $\theta_0$ and $\theta$.

Note that $\tilde{\alpha}^{(l+1)}$ is linear in $\mathbf{w}^{(l+1)}$, hence, $\frac{\partial^2 \tilde{\alpha}^{(l+1)}}{(\partial \mathbf{w}^{(l+1)})^2}$ is always a zero matrix. Hence, term $\mathcal{A} \equiv 0$. Moreover, by Theorem 4, the term $\mathcal{B}$ becomes zero, as the width $m_l$ increases to infinity. For the term $\mathcal{C}$, we note that

$$\frac{\partial^2 \tilde{\alpha}^{(l+1)}}{\partial \mathbf{w}^{(l+1)} \partial \theta^{(l)}} = \frac{1}{\sqrt{m_l}} \nabla \alpha^{(l)}.$$

Here, $\alpha^{(l)} = (\alpha_1^{(l)}, \ldots, \alpha_{m_l}^{(l)})$ is the vector of the neurons in layer $l$ and $\nabla \alpha^{(l)}$ is the matrix of the first-order derivative of the vector $\alpha^{(l)}$. Using Lemma 3, we have an upper bound on the spectral norm: $\|\nabla \alpha^{(l)}\| \leq \sqrt{C}$, with $C > 0$ being a constant. Therefore, we have

$$|\mathcal{C}| \leq \frac{1}{\sqrt{m_l}}\|\mathbf{w}^{(l+1)} - \mathbf{w}_0^{(l+1)}\|\|\nabla \alpha^{(l)}\|\|\theta^{(l)} - \theta_0^{(l)}\| \leq \frac{R^2 \sqrt{C}}{\sqrt{m_l}}. \tag{29}$$

In the limit of $m_l \to \infty$, term $\mathcal{C}$ converges to 0. Therefore, we have proved that each $(l + 1)$-th layer pre-activation $\tilde{\alpha}^{(l+1)}$ has a vanishing quadratic term in its Taylor expansion Eq.(1), and hence becomes linear, in the limit of $m_l \to \infty$.

# E   DENSENET

In a DenseNet (Huang et al., 2017), a $l$-th layer neuron takes all previous layer neurons as inputs:

$$\alpha^{(l)} = \sigma(\tilde{\alpha}^{(l)}), \quad \tilde{\alpha}^{(l)} = \frac{1}{\sqrt{m}} W^{(l)} * \mathrm{Concat}[\alpha^{(l-1)}, \ldots, \alpha^{(1)}, \mathbf{x}], \tag{30}$$

where the "Concat" function concatenates all the vector arguments into a single long vector with $m$ being the size, and $*$ is the matrix multiplication. The pre-activation can be viewed as the sum of $l$ linear functions, $\tilde{\alpha}^{(l)} = \sum_{l'=1}^{l} \tilde{\alpha}_U^{(l')}$, where each $\tilde{\alpha}_U^{(l')} := \frac{1}{\sqrt{m}} U^{(l')} \alpha^{(l')}$ has a similar form as the pre-activation in full-connected neural network Eq.(8), but with parameters $U^{(l')}$ being a subset of $W^{(l)}$. Hence, using the analysis in Section 3, in the infinite network width limit and in $O(1)$-neighborhoods of the initialization, each of the pre-activations of the DenseNet is a summation of $l \leq L$ linear functions, and hence is linear. Therefore, the DenseNet is also linear in $O(1)$-neighborhoods of the initialization.

