# OpenReview forum: "Transition to Linearity of Wide Neural Networks is an Emerging Property of Assembling Weak Models"
_ICLR.cc/2022/Conference — ICLR 2022 Spotlight_

### Official Review · Reviewer_nTFG · 2021-11-02

**Correctness:** 3
**Technical Novelty And Significance:** 2
**Empirical Novelty And Significance:** Not applicable
**Recommendation:** 6
**Confidence:** 2

**Main Review:**

Strength:

The authors managed to providing a new perspective on why the wider neural networks tend to be linear, which looks interesting and also consistent with previous results. They also extend the basic results to more complex multi-layer cases.

As this paper is out of my expertise and purely theoretical, I hope other reviewers with more background can comment more on its novelty and significance.



Weakness:

It's not clear to me how the main theoretical results compare against previous methods on linearity of neural networks. Taking Theorem 1 for example, is the convergence term s(m) faster or slower when compared with previous results? Another general comment is that a specific section on "related work" may make the connection with previous work more clear.

**Summary Of The Paper:**

This paper aims to explain the phenomenon that neural networks with infinite width tend to be linear in the neighborhood of initial optimization points, from an assemble model perspective. The key point is to prove that the quadratic term from Taylor expansion disappears when the width approaches infinity. The authors first proved in the simple two-layer case and then extend the results to deep neural networks with L layers.

**Summary Of The Review:**

This paper provides a new perspective towards an interesting observation in deep neural networks, while the connection with previous work could be stated more explicitly.

---

> ### Author Response · Authors · 2021-11-16
> **Response to Reviewer nTFG**
>
> Thank you for your positive feedback.
>
> > It's not clear to me how the main theoretical results compare against previous methods on linearity of neural networks.
>
> Our work aims to identify the fundamental principles that underlie the “transition to linearity” (or constant NTK) observed in prior works. Previous analyses on linearity of neural networks are quite mathematical, and it is not necessarily intuitively clear what underlying structures account for it. This work  clarifies these principles from the perspective of the assembly model.
>
> > Taking Theorem 1 for example, is the convergence term s(m) faster or slower when compared with previous results?
>
> For standard neural networks s(m) is the same as in previous works.
>
> > a specific section on "related work" may make the connection with previous work more clear.
>
> The paragraphs, "boosting" and "Bagging and Reservoir computing", at the end of introduction discuss the related works that similar to the concept of assembly model. We will also add a discussion of the mean field analysis of neural networks

---

### Official Review · Reviewer_rv6p · 2021-11-05

**Correctness:** 4
**Technical Novelty And Significance:** 4
**Empirical Novelty And Significance:** Not applicable
**Recommendation:** 8
**Confidence:** 3

**Main Review:**

Authors present a very interesting and novel theoretical contribution to the understanding of contemporary deep learning architectures with increasing network width. Contributions of this study neatly follows the line of investigations presented by [Lee et al. 2019], and [Liu et al., 2020a]. The paper is very clearly written, and I find this work to be of interest for presentation at ICLR.

The paper provides a rigorous theoretical understanding of the phenomenon, while it naturally lacks empirical contributions and demonstrations overall. Some comments that the authors can discuss/include:
- What does the results tell us about the impact of common practices in network regularization (e.g., with dropout)?
- Any discussions on how can the performed assembly analysis extend to residual networks (e.g., WideResNets)?
- Is it possible to perform any synthetic data based empirical/numerical validation (e.g., for wide neural networks with linear output and a bottleneck) of the finding on the orthogonality of the gradient directions for neurons within the same layer of a network with large width?

**Summary Of The Paper:**

Authors present a theoretical understanding for the "transition to linearity" phenomenon of wide neural networks with linear output layer under gradient descent. Presented line of thought develops from the perspective where one considers a neural network as a recursively-built assembly of sub-models which correspond to individual neurons from the preceding layers. Authors prove that the transition to linearity phenomenon when the number of sub-models converge to infinity (i.e., the network becomes wider) is a result of assembling weak sub-models that do not dominate the assembly from network initialization.


**Summary Of The Review:**

The paper is well written and presents a novel theoretical contribution to the understanding of the recently highlighted phenomenon on how wide neural network architectures with linear output layer inherently evolve as linear models. I find this work to be of good quality in terms of theoretical contribution, whereas on the other side empirically weak. My rating would be to accept the paper for presentation at ICLR, as long as the authors can present some numerical verifications or discussions on the how to make an understanding of these results for a broader audience.

---

> ### Author Response · Authors · 2021-11-16
> **Response to Reviewer rv6p**
>
> We thank the reviewer for the positive feedback.
>
> > What does the results tell us about the impact of common practices in network regularization (e.g., with dropout)?
>
> We think our results generalize to neural networks with dropout without significant modifications. Here is a sketch of the argument.
>
> Dropout basically means that, when we are constructing a $l+1$ layer pre-activation neuron, we use a subset $\mathcal{S}$ of randomly selected $l$-layer post-activation neurons as the "active" sub-models, rather than the whole set as in the case without dropout. As shown in the paper, the  $l$-layer post-activation neurons become independent to each other in the infinite width limit (or approximately independent for large width). This also holds true for the subset $\mathcal{S}$. Hence, all our analysis applies, as long as the size of this subset is also large. This is guaranteed as the dropout rate is usually a constant.
>
> > Any discussions on how can the performed assembly analysis extend to residual networks (e.g., WideResNets)?
>
> The $O(1)$-neighborhood linearity also holds for wide ResNets, as shown in the related works [Du et al. 2019], [Liu, et al. 2020].
> While we believe our assembly point of view also applies to wide ResNet architectures, our analysis would need to be extended to accommodate cross-layer connections in the ResNet setting. We leave that for future work.
>
> > Is it possible to perform any synthetic data based empirical/numerical validation (e.g., for wide neural networks with linear output and a bottleneck) of the finding on the orthogonality of the gradient directions for neurons within the same layer of a network with large width?
>
> Thanks for this suggestion.
>
> To confirm the orthogonality numerically, we have run a set of  experiments, with a 4 layer fully connected neural network on 20 data points from CIFAR-2. Specifically, given two neurons $\tilde{\alpha}_i$ and $\tilde{\alpha}_j$, We compute the cosine angle $\cos\theta$ between the gradients $\nabla\tilde{\alpha}_i$ and $\nabla\tilde{\alpha}_j$ as $\cos\theta=<\nabla\tilde{\alpha}_i,\nabla\tilde{\alpha}_j>/\|\nabla\tilde{\alpha}_i\|\|\nabla\tilde{\alpha}_j\|$. We present the average of $\cos\theta$ over all pairs of $(i,j)$ with $i\ne j$ and over all data samples to measure the orthogonality. The following results show that as the network width $m$ increases, the average $\cos\theta$ decreases towards zero. This confirms our theoretical result that in wider networks the gradients of neurons become more orthogonal.
>
>   m=20,         average $\cos \theta$ = 0.1454
>
>   m=50,         average $\cos \theta$ = 0.0880
>
>   m=100,        average $\cos \theta$ = 0.0753
>
>   m=500,        average $\cos \theta$ = 0.0264
>
>   m=1000,       average $\cos \theta$ = 0.0185
>
> We also verified this in a bottleneck network of the following architecture: 4 layer fully connected neural network, with 1st layer and 3rd layer with large width $m=1000$ and the 2nd layer as the bottleneck layer with width $m_b$.
>
>   $m_b$=1,          average $\cos \theta$ = 1
>
>   $m_b$=50,         average $\cos \theta$ = 0.0374
>
>   $m_b$=100,        average $\cos \theta$ = 0.0280
>
>   $m_b$=1000,       average $\cos \theta$ = 0.0185
>
> We plan to add some more extensive experiments in the revision.

---

> > ### Comment · Reviewer_rv6p · 2021-11-22
> > **Thanks for responses**
> >
> > Thanks to the authors for their responses and additional analyses. I suggest that authors include these empirical validations and provided discussions in their MS revisions, which are not yet incorporated by edits on the MS so far. I will keep my rating (suggesting an acceptance) as it is.

---

> > > ### Author Response · Authors · 2021-11-22
> > > **reply**
> > >
> > > We thank the reviewer for the response and the suggestion on these empirical validations.
> > >
> > > We think that these empirical experiments should be an improvement of the article. Now, we are conducting more such experiments in various settings. We plan to add a detailed discussion about these numerical verifications on a range of experimental settings. These updates will appear in the camera ready.

---

### Official Review · Reviewer_r126 · 2021-11-07

**Correctness:** 4
**Technical Novelty And Significance:** 3
**Empirical Novelty And Significance:** Not applicable
**Recommendation:** 8
**Confidence:** 3

**Main Review:**

The paper is very well written, with proper style and easy to follow. The findings look interesting and expand the work of other manuscripts. My main concerns are as follow:

1. The introduction mainly focuses on the proposed new perspective (assembly model) without much discussion of the benefits of it. This could also be solved at the end of the manuscript with a discussion and limitations section that explains better which future work does this work provide.

2. The related work at the end of the introduction is a bit short. I think further discussion of the similarities and limitations of the proposed perspective with works like Liu et al. 2020a and Lee et al. 2019 is needed. I also miss some references to works that analyse the mean field gradient descent dynamics of training neural networks in the large-width limit and the relation to the submitted manuscript.

3. The key finding of this manuscript is "[...], when the network width is large, the neurons within the same layer become essentially independent to each other in the sense that their gradient directions are orthogonal in the parameter space.". This is linked to assembling independent sub-models to obtain the O(1)-neighbourhood linearity. It seems like the proposed perspective relies directly onto Assumption 1. However, my impression is that if Assumption 1 is not there, other properties could emerge that explain the transition to linearity. The authors start by fixing the property of the neural network being an assembly model and work from there, but does that imply that there are no other structural properties that could explain the transition to linearity on wide neural networks? Is it guaranteed that the constructed assembly model is not dominated by one or a minor group of sub-models?

Minor typos:
page 3: Further more --> Furthermore
pages 4, 5, 7 and 8: the the --> the


**Summary Of The Paper:**

This paper aims to explore what structural properties account for transition to linearity when a neural network becomes wider. The authors propose a new perspective where the neural network is considered as a multi-level assembly model. Each hidden layer can be viewed as an assembly model built up from a set of independent sub-models. Furthermore, they assume that the proposed assembly model is not dominated by a single or minor group of sub-models. This leads to neurons within the same hidden layer being independent when the network width grows to infinity, and to show that the network is linear in an O(1)-neighbourhood around the initialization.

**Summary Of The Review:**

The paper looks solid and well written, although the contribution and usefulness of the findings is not made totally clear. I am not an expert on this field, thus some more relevant related work might be missing. Most of the mathematical formulation seems correct, although I might have missed some details. My main concern is the way in which the findings are presented seem to point as to the proposed perspective (assembly model) being one of the explanations under Assumption 1, but not necessarily the only one, or valid when said assumption is not true.

---

> ### Author Response · Authors · 2021-11-16
> **Response to Reviewer r126**
>
> We thank the reviewer for the positive feedback. We address your concerns below.
>
> > 1. The introduction mainly focuses on the proposed new perspective (assembly model) without much discussion of the benefits of it. This could also be solved at the end of the manuscript with a discussion and limitations section that explains better which future work does this work provide.
>
> Thanks for the suggestion, we will extend the discussion in the next revision.
>
> > 2. I also miss some references to works that analyse the mean field gradient descent dynamics of training neural networks in the large-width limit and the relation to the submitted manuscript.
>
> In this work we concentrate on the NTK setting, which is different from the mean field setting. We will add a discussion of the mean field in the revision.
>
> >  3. However, my impression is that if Assumption 1 is not there, other properties could emerge that explain the transition to linearity. The authors start by fixing the property of the neural network being an assembly model and work from there, but does that imply that there are no other structural properties that could explain the transition to linearity on wide neural networks? Is it guaranteed that the constructed assembly model is not dominated by one or a minor group of sub-models?
> My main concern is the way in which the findings are presented seem to point as to the proposed perspective (assembly model) being one of the explanations under Assumption 1, but not necessarily the only one, or valid when said assumption is not true.
>
> Assumption 1 is generally true for wide neural networks. This is because the neurons within the same layer are constructed on an equal footing, only with different random seeds that generate linear combination coefficients. More specifically, in Eq.(8) of the submission, the difference is on the rows of the matrix $W$, which follow the same distribution. As a result, the values of the neurons in the same layer are at the same order, typically $\Theta(1)$, and satisfies assumption 1.
>
> We believe that  Assumption 1, which makes sure no dominating sub-models, is also necessary to some degree, if a transition to linearity to occur. Consider, for example a simple two-layer neural network  in Eq.(7). If one of the sub-models, say $\sigma(w_i^Tx)$ dominates, the neural network cannot be close to linear, unless the activation function $\sigma$ is linear itself.

---

> > ### Comment · Reviewer_r126 · 2021-11-22
> > **Thanks for the responses**
> >
> > Thanks to the authors for their responses, and the discussion with the other reviewers. Most of my concerns have been addressed and therefore I would be maintaining my rating if the proposed changes by the authors are applied to the manuscript. Furthermore, I find that the empirical experiments to be added (see rv6p thread) look quite interesting for the improvement of the article. I would like, though, that the architecture used be properly explained in detail.

---

> > > ### Author Response · Authors · 2021-11-22
> > > **Reply**
> > >
> > > We thank the reviewer for the response.
> > >
> > > We agree that these empirical experiments should be an improvement of the article. We are conducting more such experiments and plan to add a detailed discussion (including experimental setup) about these numerical verifications, together with the other promised changes. These updates will appear in the camera ready.

---

### Decision · Program_Chairs · 2022-01-20

**Decision:**

Accept (Spotlight)

**Comment:**

The authors provide in this manuscript a theoretical analysis to explain why deep neural networks become linear in the neighbourhood of the initial optimisation point as their width tends to infinity. They approach this question by viewing the network as a multi-level assembly model.

All reviewers agree that this is an interesting, novel, and relevant study. The paper is very well-written.

Initially, a weak point raised by a reviewer was that an empirical evaluation of the theory was missing. The authors addressed this issue in a satisfactory manner in their response.

In conclusion, this is a strong contribution worth publication.